# Safety and efficacy of bempedoic acid among patients with statin intolerance and those without: A meta-analysis and a systematic randomized controlled trial review

Yi Li *, Hongyu Gao, Jinghui Zhao, Liqing Ma, Dan Hu

Department of Cardiology, No 1 Hospital Of Baoding, Baoding, Hebei, China

* liyi251668@163.com

## Abstract

### Objective

Bempedoic acid, an innovative oral medication, has garnered significant interest in recent times due to its potential as a therapeutic intervention for hypercholesterolemia. Nonetheless, the outcomes of the initial investigations might have been more definitive and coherent. Our objective was to perform a quantitative meta-analysis in order to evaluate bempedoic acid's safety and effectiveness.

### Methods

A search was conducted on ClinicalTrials.gov, and PubMed from the time of inception until September 28, 2023. Randomized controlled trials comparing the safety and efficacy of bempedoic acid among patients with statin intolerance and those without were included in our analysis. The trial outcomes were summarized using a random effects model and were provided as mean differences or odds ratios (ORs) with a confidence interval of 95%. Additionally, trial heterogeneity and the possibility of bias were evaluated and investigated.

### Results

Bempedoic acid treatment reduced low-density lipoprotein cholesterol levels more than placebo (mean difference -2.97%, 95% CI -5.89% to -0.05%), according to a pooled analysis of 16 eligible trials. The risk of death (OR 1.18, 95% CI 0.70 to 1.98) and muscle-associated occurrences (OR 1.00, 95% CI 0.77 to 1.31) was not impacted by bempedoic acid. In contrast, discontinuation of treatment was more frequently caused by adverse events in the bempedoic acid group (OR 1.13, 95% CI 1.01 to 1.27).

### Conclusions

In patients with statin intolerance as well as those without, bempedoic acid is a safe and efficacious lipid-lowering agent, according to findings from randomized controlled trials.

**Data Availability Statement:** All relevant data are within the manuscript and its Supporting Information files.

**Funding:** The author(s) received no specific funding for this work.

**Competing interests:** The authors have declared that no competing interests exist.

## Introduction

Cardiovascular diseases (CVDs) remain the primary contributor to mortality and disability in developed nations [1]. Atherosclerotic cardiovascular diseases (CVDs) risk may be mitigated by a lifetime decrease in low-density lipoprotein cholesterol (LDL-C), according to Mendelian randomization studies [2]. Consistent evidence from controlled clinical trials established a correlation between LDL-C reduction and decreased cardiovascular (CV) risk [3]. As a result, lipid-lowering treatment emerged as a fundamental approach to reducing CV risk.

As an innovative oral medication, bempedoic acid (ETC-1002) is administered once daily for treating hyperlipidemia. It cannot be completely active in skeletal muscle; rather, it must be activated in the liver. Meanwhile, the activated state impedes ATP-citrate lyase, an enzyme situated HMG-CoA reductase upstream. This results in an increase in the expression of hepatic LDL receptors, an improvement in LDL particle clearance, and a reduction in the levels of circulating LDL-C [4–6]. Bempedoic acid functions analogously to statins and potentially exhibits very low myotoxicity.

Nevertheless, the prospective advantages of bempedoic acid for evading cardiovascular events remain unproven in ongoing clinical trials [1, 7]. As a result, the benefits of bempedoic acid are currently restricted. A systematic review and meta-analysis were undertaken in order to assess the possible efficacy and safety profile of bempedoic acid in preventing cardiovascular events.

## Methodology

The methods utilized in this meta-analysis were in strict adherence to the typically utilized reporting forms for systematic reviews and meta-analyses. Two evaluators (YL and HYG) extracted data, evaluated the risk of bias, and obtained studies for eligibility in an independent manner. The two evaluators achieved a consensus after discussing process inconsistencies. When applicable, a third reviewer (LQM) was consulted.

### Literature retrieval

Up until and including September 28, 2023, we perused PubMed and ClinicalTrials.gov in the case of randomized controlled trials (RCTs) examining the safety and efficacy of bempedoic acid. For database searches, the subsequent medical subject headings and keywords had been employed: (bempedoic acid OR nilemdo OR nexletol OR ETC-1002 OR ESP-55016) AND (randomized controlled trial OR placebo). We refrained from imposing any further restrictions on the constrained pertinent records within this approach. In addition to the reference lists of possible eligible papers and prior systematic reviews, we conducted a manual search for any additional eligible studies that were not identified during the preliminary database literature perusal.

### Selection of studies

The inclusion criteria were specified in detail as follows: (a) Population: patients, whether or not they have a statin intolerance; (b) Intervention: bempedoic acid administered at any dose; (c) Control: a placebo-controlled group; and (d) Outcomes: treatment-related efficacy and safety characteristics. Excluded from consideration were correspondences, proceedings, editorials, expert opinions, evaluations lacking first-hand information, and nonhuman papers. Additionally, phase 1 clinical trials and duplicate publications were not considered, and there was no comparison between bempedoic acid and placebo groups. A more extensive iteration was incorporated to account for duplicate research.

## Data extraction

Two examiners (YL and HYG) independently obtained the subsequent study parameters: author, year, study design, duration, population, sample, age, female proportion, and results. The main efficacy endpoint for the sixteen studies that were included was the change (in %) in LDL-C from baseline to at most 40.6 months. To evaluate the effectiveness of bempedoic acid, we obtain data at the respective time point, which comprised high-density lipoprotein cholesterol (HDL-C), LDL-C, non-HDL-C, apolipoprotein B (apoB), total cholesterol (TC), triglycerides, and high-sensitivity C-reactive protein(hsCRP). With regard to safety, we focused on adverse events leading to discontinuation, muscle-related events, gout events, death events, diarrhea events, constipation events, hepatic enzyme increased events, renal impairment events, headache events, myocardial infarction events, stroke events, and new-onset or worsening diabetes.

## Risk of bias (quality) assessment

Bias risk in valid RCTs was assessed independently by two reviewers (YL and HYG) utilizing the Cochrane Collaboration tool. Incomplete outcome data (attrition bias), selective reporting (reporting bias), allocation concealment (selection bias), random sequence generation (selection bias), outcome assessment blinding (detection bias), and other biases (including whether baselines were comparable) were taken into account.

## Statistical examination

After extraction from eligible trials, continuous data were aggregated as mean difference (MD) with standard deviations and dichotomous data were pooled as OR with 95% CI. As stated in the Cochrane Handbook for Systematic Review of Interventions, the standard error or 95% CI was utilized to compute lacking standard deviations. Qualitative descriptions of outcomes that evaded meta-analysis were provided in the text. Change in LDL-C (in %) was the principal efficacy endpoint in the trials that were included. Data was obtained at the designated time point in order to evaluate the effectiveness of bempedoic acid.

Cochran's Q test and Higgins I2 statistics were employed to assess the presence of prospective statistical heterogeneity between trials. A meta-analysis will be conducted utilizing a model with random effects. Subgroup analyses were performed in order to assess heterogeneity and identify potential confounding variables, such as variations in the patients enrolled. The "trim and fill" approach of sensitivity analysis was implemented in order to identify possible heterogeneity sources. In order to assess bias in publication, the Egger's and Harbord's tests were implemented.

All analyses were performed using Review Manager (RevMan version 5.4 for windows) and STATA.17 (Stata Corp., College Station, Texas, USA). A two-tailed p-value less than 0.05 was considered significant.

# Results

## Characteristics of included studies

Out of the 126 studies that were obtained through the search strategy, 49 were evaluated through a comprehensive perusal of the full texts. Eventually, sixteen randomized studies that satisfied the inclusion parameters were part of this meta-analysis. A summary of the selection procedure is provided in Fig 1. A summary of the attributes of the trials that were included can be found in Table 1. The combined number of enrolled patients amounted to 18469, of which 8624 were administered a placebo and 9845 were administered bempedoic acid. Except for the John Rubino et al. trial [8], the John Rubino et al. trial [9], the Narendra D. Lalwani et al. trial

## PRISMA 2020 flow diagram for new systematic reviews which included searches of databases and registers only

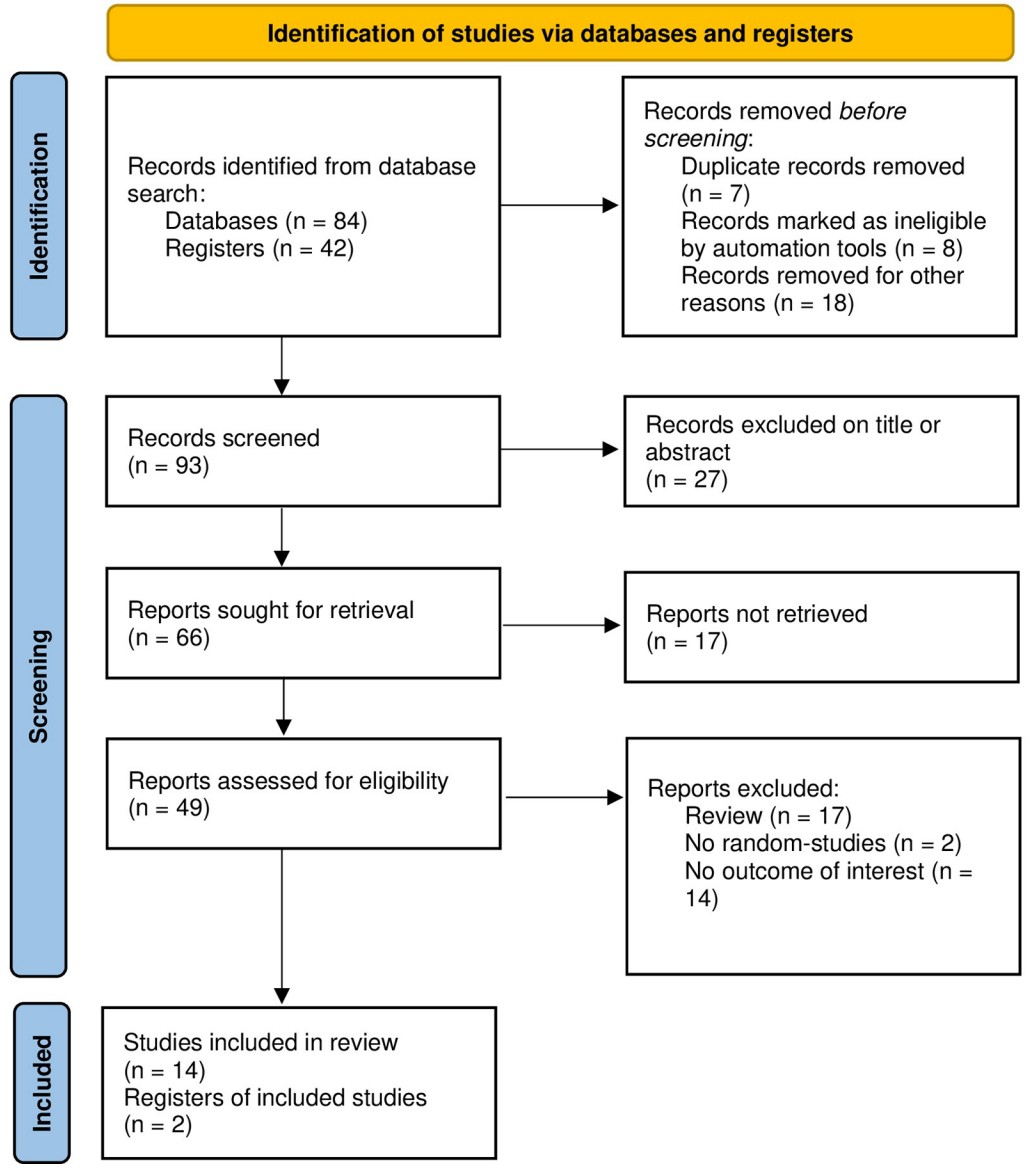

**Fig 1. Flow diagram to show study search process.**

[10], the Paul D. Thompson et al. trial [11], and the Maria J. Gutierrez et al. trial [12], all studies enrolled more than 100 patients. The S.E. Nissen et al. trial [13], the Ulrich Laufs et al. trial [14], the Christie M. Ballantyne et al. trial [15], and the Paul D. Thompson et al. trial [11] were conducted on patients with statin intolerance. In contrast, other studies were conducted on patients who did not have statin intolerance.

## Overall efficacy

Fig 2 illustrates the outcomes obtained by integrating the mean difference of LDL-C using a random-effects model. Inclusion in the studies revealed a range of lipid-lowering effectiveness.

**Table 1. Basic features of the included studies.**

| Author | Year | Study | Population | Number | Intervention | control | Outcome | Treatment duration | Age(years), mean ± SD | Female, n (%) |
|---|---|---|---|---|---|---|---|---|---|---|
| S.E. Nissen | 2023 | RCT | patients who were unable or unwilling to take statins or were at high risk for cardiovascular disease | 13,970 | bempedoic acid | placebo | component composite of major adverse cardiovascular events | 40.6 months | 65.5±9.0 | 6740 (48.2) |
| John Rubino | 2021 | RCT | patients with hypercholesterolemia | 58 | bempedoic acid | placebo | percent change in LDL-C | 2months | 60.14±10.44 | 36(62.1) |
| John Rubino | 2021 | RCT | patients with hypercholesterolemia | 63 | bempedoic acid,ezetimibe, atorvastat | placebo | percent change from in LDL-C | 6weeks | 61.2±11.0 | 40(63.5) |
| Christie M Ballantyne | 2020 | RCT | patients with hypercholesterolemia and high CVD risk | 301 | bempedoic acid,ezetimibe | placebo | percentage change in LDL-C | 12weeks | 64.3±9.50 | 152 (50.5) |
| Kausik K. Ray | 2019 | RCT | patients with atherosclerotic cardiovascular disease, heterozygous familial hypercholesterolemia, or both | 2230 | bempedoic acid | placebo | percentage change in LDL-C | 12 to 52 weeks | 66.1±8.9 | 602 (27.0) |
| Ulrich Laufs | 2019 | RCT | patients with hypercholesterolemia and statin Intolerance | 345 | bempedoic acid | placebo | percent change in LDL-C | 24weeks | 65.2±9.5 | 194 (56.2) |
| Narendra D. Lalwani | 2019 | RCT | patients with hypercholesterolemia | 64 | bempedoic acid | placebo | the LDL-C lowering efficacy | 4weeks | 58±9.3 | 31(48.4) |
| Anne C. Goldberg | 2019 | RCT | patients with atherosclerotic cardiovascular disease, heterozygous familial hypercholesterolemia, or both | 779 | bempedoic acid | placebo | percent change in LDL-C | 52weeks | 64.3±8.8 | 283 (36.3) |
| Christie M. Ballantyne | 2018 | RCT | patients with a history of statin intolerance | 269 | bempedoic acid | placebo | percent change in LDL-C | 12weeks | 63.8±10.9 | 165 (61.3) |
| Paul D. Thompson | 2016 | RCT | hypercholesterolemic patients | 348 | bempedoic acid | placebo | percent change in LDL-C | 12weeks | 59.9±9.6 | 265 (79.1) |
| Christie M. Ballantyne | 2016 | RCT | hypercholesterolemic Patients | 133 | bempedoic acid | placebo | percent change in LDL-C | 12weeks | 57.3±9.7 | 79(59.4) |
| Paul D. Thompson | 2015 | RCT | hypercholesterolemia in patients with statin intolerance | 56 | bempedoic acid | placebo | percent change in LDL-C | 8weeks | 62.6±6.4 | 28 (50.0%) |
| Maria J. Gutierrez | 2014 | RCT | patients with type 2 diabetes mellitus and hypercholesterolemia | 60 | bempedoic acid | placebo | the lipid-altering effects | 4weeks | 55.7±8.5 | 23(38.3) |
| Christie M. Ballantyne | 2013 | RCT | patients with hypercholesterolemia | 177 | bempedoic acid | placebo | changes in LDL-C | 12weeks | 57.7±9.5 | 79(44.6) |
| NCT03531905 | 2020 | RCT | patients with type 2 diabetes (T2D) and elevated LDL-C | 242 | bempedoic acid | placebo | LDL-C lowering | 12weeks | 61.4±8.4 | 117 (48.3) |
| NCT02178098 | 2023 | RCT | participants with hypercholesterolemia and hypertension | 143 | bempedoic acid | placebo | the efficacy and safety of ETC-1002 | 6weeks | 55.6±8.4 | 61(42.7) |

Generally, bempedoic acid reduced LDL-C levels more significantly than placebo (MD -2.97%, 95% CI -5.89% to -0.05%; p = 0.046; $I^2$ = 99.9%) (Fig 2). Significant reductions were also observed for non-HDL-C (MD -1.30%, 95% CI -1.59% to -1.02%; p = 0.000; $I^2$ = 92.2%), TC (MD -1.32%, 95% CI -1.60% to -1.05%; p = 0.000; $I^2$ = 91.4%), apoB (MD -1.17%, 95% CI -1.44% to -0.90%; p = 0.000; $I^2$ = 90.7%), and HDL-C levels (MD -0.34%, 95% CI -0.46% to -0.23%; p = 0.000; $I^2$ = 11.6%) (Figs 2 and 3). Meanwhile, we found no statistical difference between the bempedoic acid and placebo groups in hsCRP (MD -1.32%, 95% CI -3.78% to

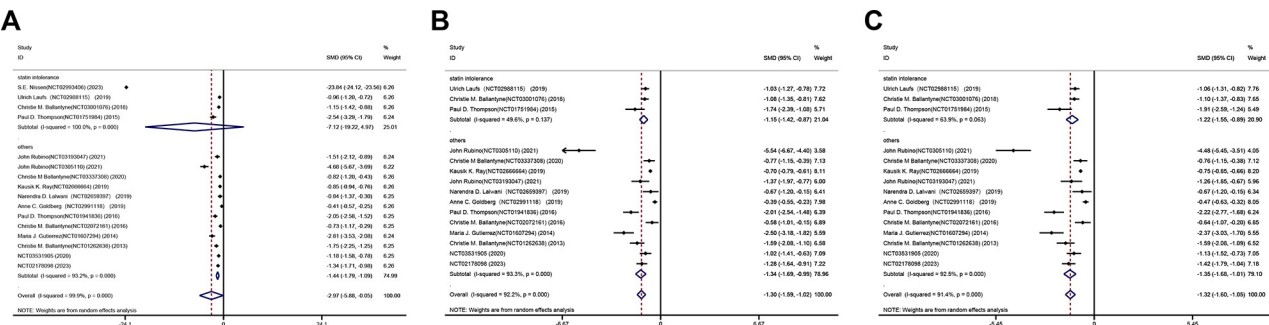

**Fig 2.** Forest plot displaying mean difference and 95% confidence intervals for the effect of bempedoic acid on plasma levels of low-density lipoprotein cholesterol(A),non-high-density lipoprotein cholesterol(B),and total cholesterol(C).

1.15%; p = 0.295; I$^2$ = 99.9%) and triglycerides (MD 0.00%, 95% CI -0.25% to 0.25%; p = 0.982; I$^2$ = 76.3%) (Fig 3).

Subgroup analyses yielded comprehensive results that led us to the conclusion that the heterogeneity source could be more adequately elucidated by the various enrolled patients. The included studies were categorized into two: (1) subgroup 1 (n = 4): patients with statin

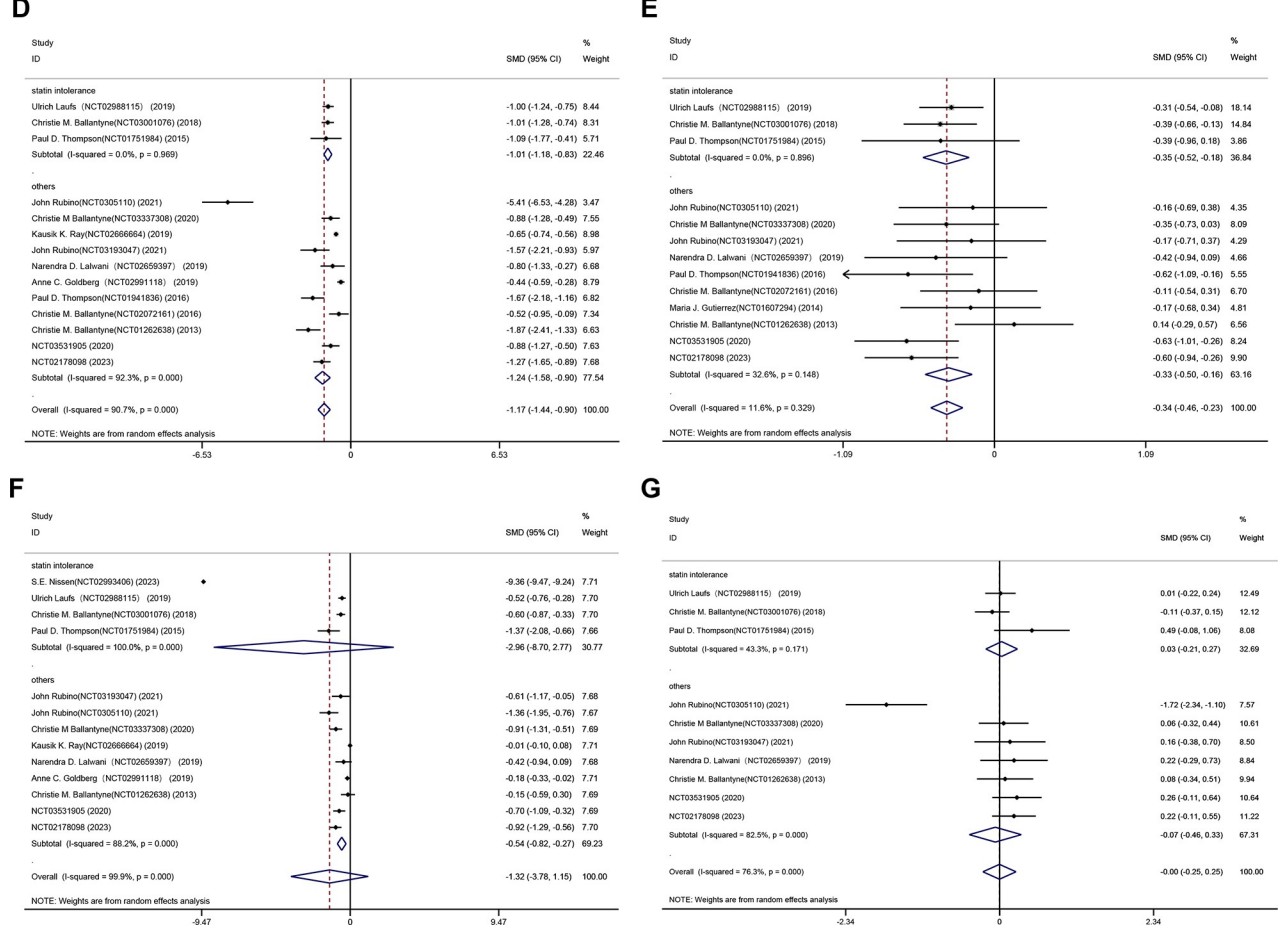

**Fig 3.** Forest plot displaying mean difference and 95% confidence intervals for the effect of bempedoic acid on plasma levels of apolipoprotein B(D), high-density lipoprotein cholesterol(E),high-sensitivity C-reactive protein(F) and triglycerides(G).

intolerance [11, 13–15]; and (2) subgroup 2 (n = 12): others (NCT03531905, NCT02178098) [8–10, 12, 16–21]. A subgroup analysis pertaining to the percent change in LDL-C indicated that background therapy without any stains resulted in no significant reduction in LDL-C (P = 0.248; Fig 2). No percent change in hsCRP was observed across the 18059 participants of 13 trials. In terms of hsCRP reduction, background therapy was more beneficial for those without stain intolerance, according to subgroup analysis of hsCRP percent change (P = 0.000; Fig 3).The levels of non-HDL-C, TC, apoB, and HDL-C were notably reduced in the aforementioned subgroups (Figs 2 and 3).

## Overall safety

The joint OR of representative adverse occurrences for the placebo and bempedoic acid groups is illustrated in Figs 4–6. Additionally, death (OR 1.18, 95% CI 0.70 to 1.98; p = 0.533; I$^2$ = 13.6%), muscle-related adverse (OR 1.00, 95% CI 0.77 to 1.31; p = 0.992; I$^2$ = 39.2%), stroke (OR 0.86, 95% CI 0.69 to 1.08; p = 0.195; I$^2$ = 0.0%) and new-onset or worsening diabetes events (OR 0.82, 95% CI 0.63 to 1.06; p = 0.133; I$^2$ = 29.8%) (Fig 4). However, certain safety consequences were noteworthy, including adverse events that led to discontinuation (OR 1.13, 95% CI 1.01 to 1.27; p = 0.039; I$^2$ = 1.7%), gout (OR 1.76, 95% CI 1.03 to 3.00; p = 0.038; I$^2$ = 15.5%), increased hepatic enzyme (OR 1.55, 95% CI 1.30 to 1.85; p = 0.000; I$^2$ = 0.0%), renal impairment (OR 1.38, 95% CI 1.24 to 1.55; p = 0.000; I$^2$ = 0.0%), and myocardial infarction events (OR 0.76, 95% CI 0.65 to 0.90; p = 0.001; I$^2$ = 0.0%) (Fig 5). Diarrhea, constipation, and headache, among other adverse effects, were comparable between the two groups (Fig 6).

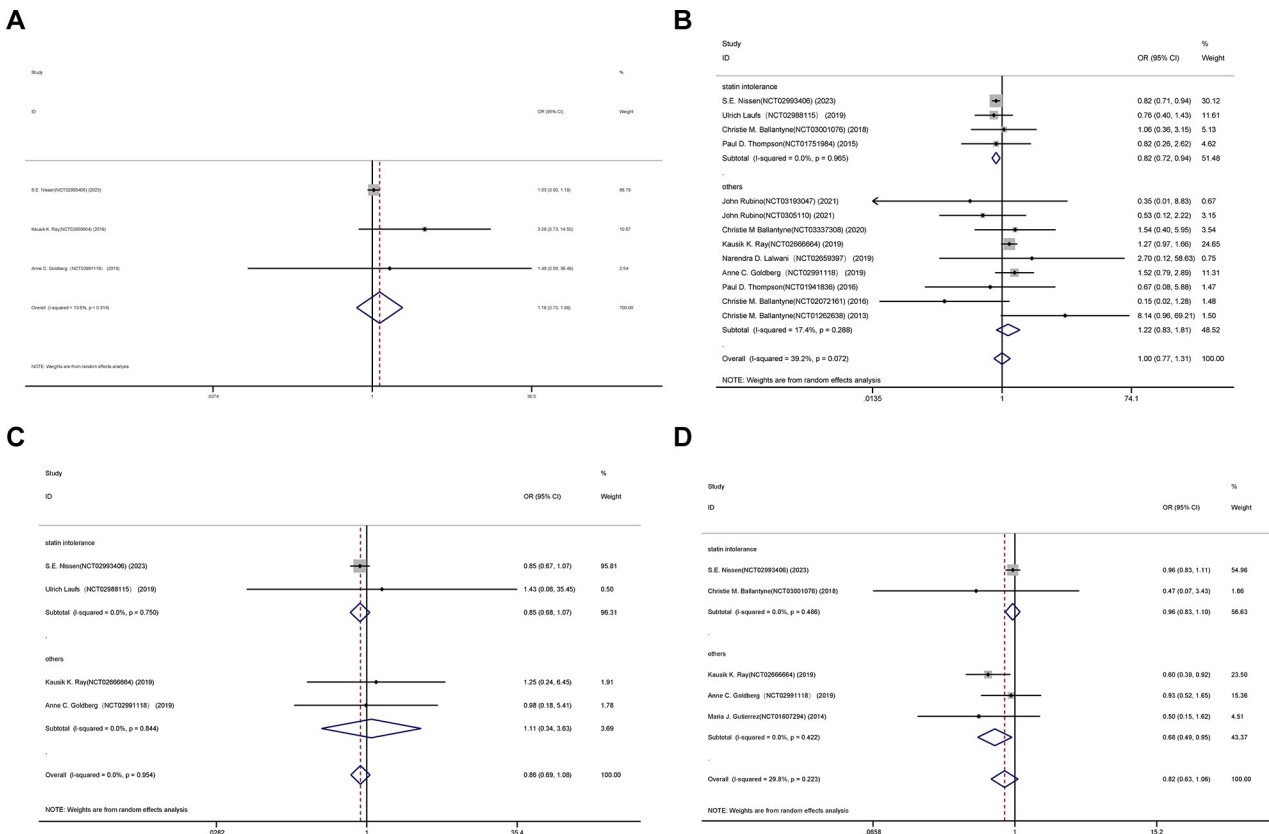

**Fig 4.** Incidence of death event(A), muscle-related adverse event(B), stroke event(C) and new-onset or worsening diabetes events(D) during treatment with bempedoic acid as compared to control treatment group.

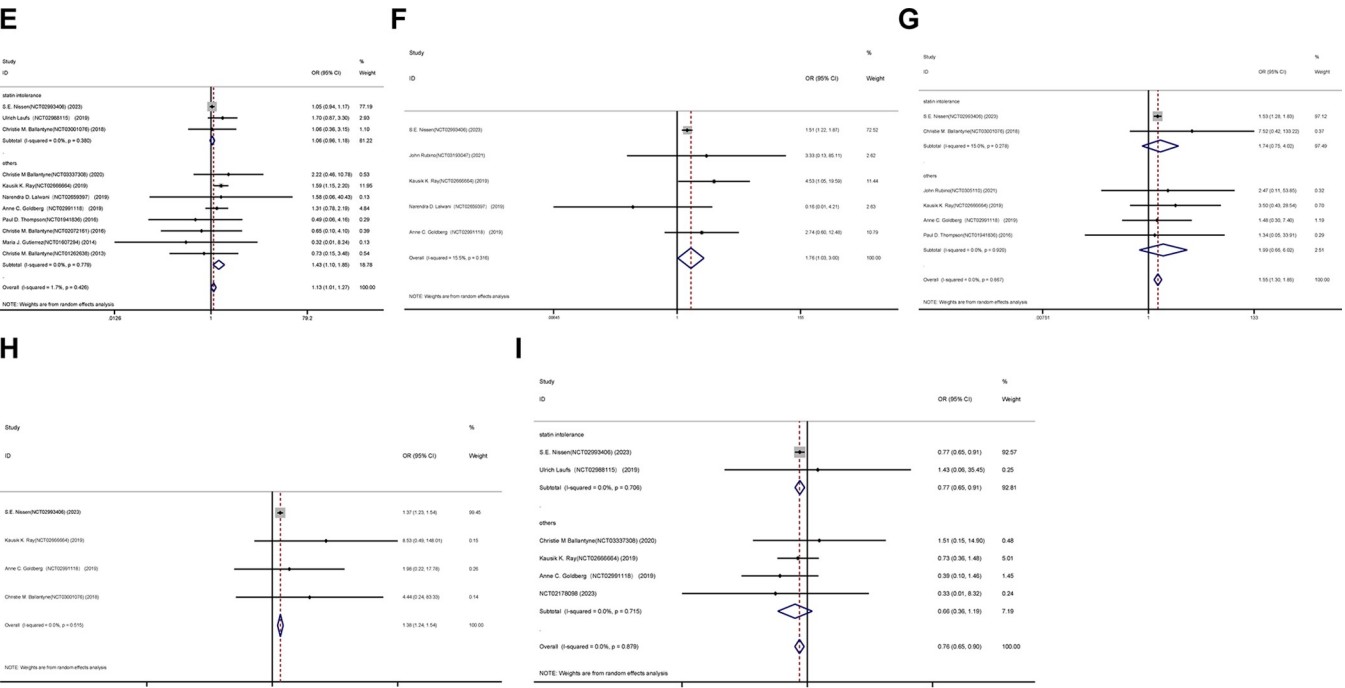

**Fig 5.** Incidence of discontinuation(E),gout(F),increased hepatic enzyme(G), renal impairment(H), and myocardial infarction events(I) during treatment with bempedoic acid as compared to control treatment group.

## Publication bias

Overall adverse events exhibited non-obvious publication bias, as demonstrated by Harbord's experiments. Publication bias was identified in TC (Egger's test p = 0.002), apoB levels (Egger's test p = 0.003), and non-HDL-C levels (Egger's test p = 0.002). To address the influence of publication bias, the researchers employed the "trim and fill" technique developed by Duval and Tweedie (S1 Fig).

## Discussion

The effectiveness and safety of bempedoic acid in patients with statin intolerance and those without were assessed in this meta-analysis. In comparison to placebo, bempedoic acid significantly decreased LDL-C, non-HDL-C, apoB, TC, and HDL-C levels, but had no discernible effect on triglycerides or hsCRP. In the subgroup of LDL-C, there is no noticeable decrease in patients with statin intolerance. However, patients without statin intolerance show contrary results. Regarding four studies [11, 13–15], they discovered a decreased level in patients with statin intolerance. According to the findings of Ulrich Laufs [3], the more substantial decrease in LDL-C in the absence of a background statin may be attributable to a shared inhibition pathway between bempedoic acid and statins, with bempedoic acid inhibiting cholesterol synthesis upstream of statins.A variation in the utilization of additional background lipid-lowering therapies is an additional improbable factor that could account for the disparity in LDL-C reduction between the two populations. Further research remains to be discovered in the future. In a theoretical framework, bempedoic acid functions in a manner analogous to that of statins, impeding the biosynthesis of ACLY-dependent cholesterol one step prior to HMG-CoA reductase. Evidently, observational studies tend to exaggerate the adverse effects of statins, and blinded RCTs fail to demonstrate any difference, a phenomenon referred to as the

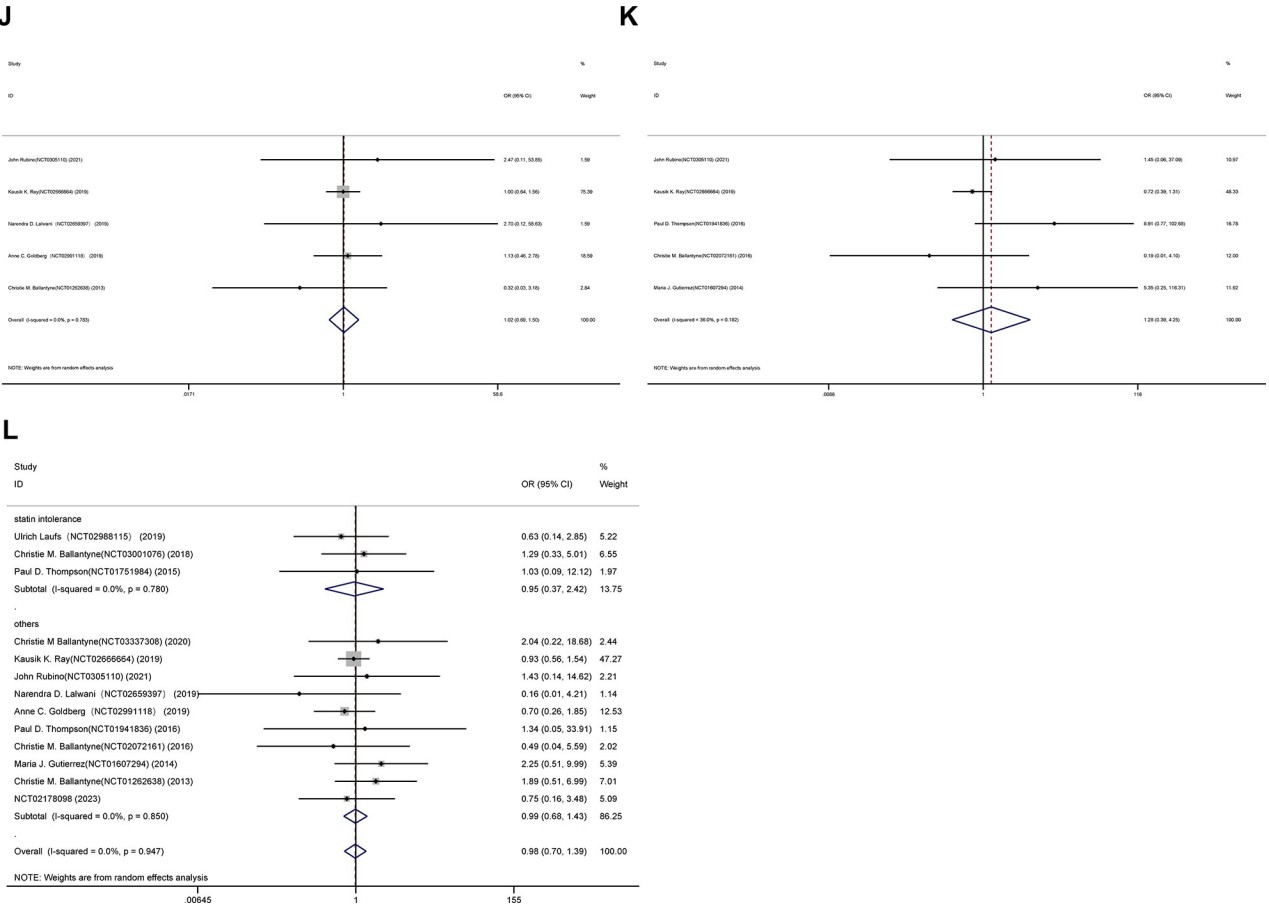

**Fig 6.** Incidence of diarrhea(J), constipation(K), and headache(L) during treatment with bempedoic acid as compared to control treatment group.

"nocebo effect" [3]. We are intrigued to observe the outcomes of empirical investigations involving bempedoic acid. Sustained reductions in hsCRP levels are possible in patients without statin intolerance. By modulating lipids and inflammation, bempedoic acid may be able to decrease the risk of cardiovascular disease, as demonstrated by the decrease. The "trim and fill" method developed by Duval and Tweedie was employed to mitigate the influence of publication bias. After using the "trim and fill" method, the results of apoB, non-HDL-C, and TC levels remained statistically significant, with no reversal. The combined results remain robust.

Safety-wise, there was no incidence of adverse events that were more severe than placebo during bempedoic acid therapy; the majority of adverse events were modest to moderate in nature. Regarding mortality, adverse muscle-related, stroke, and new-onset or worsening diabetes events, and additional factors, no statistically significant difference was observed. However, certain adverse effects were identified as reasons for treatment discontinuation. These adverse effects included events leading to treatment discontinuation, gout, elevated hepatic enzymes, renal impairment, and myocardial infarction events. These findings underscore the importance of considering these potential adverse effects when assessing the overall risk-benefit profile of bempedoic acid. Discontinuation incidences like adverse musculoskeletal consequences, were comparable to those observed with a placebo, according to the representative study by S.E. Nissen [13]. Moreover, the increased prevalence of gout in the experimental group may have been attributed to the potential for bempedoic acid to compete for renal transporters responsible for

uric acid excretion, resulting in elevated uric acid [13]. As for hepatic enzyme increased events, subgroup analyses report no apparent difference between the two groups, and renal events include four studies; more trials may need to be involved. In the subtype of myocardial infarction events, there is no apparent increase in patients without statin intolerance, while there is an apparent increase in patients with statin intolerance. As Ulrich Laufs [14] reported, individuals who are unable to tolerate statins have an elevated risk of developing cardiovascular disease as a result of chronic atherogenic lipid accumulation. Nonstatin alternatives, including ezetimibe, bile acid sequestrants, and fibrates, exhibit a comparatively diminished impact on LDL-C reduction compared to statins. Consequently, they might prove inadequate in independently mitigating the risk of cardiovascular events and lowering LDL-C.

When assessing the potential benefits of bempedoic acid, it is essential to consider its comparison with other drugs, especially statins. For instance, by reducing muscle-related symptoms, bempedoic acid may enhance patient adherence to lipid-lowering therapy, which could be critical for the clinical outcomes of long-term management of patients with high cholesterol. On the other hand, the occurrence of more adverse effects may impact patients' quality of life and potentially necessitate additional medical interventions. Additionally, it is important to comprehensively consider both its advantages and potential drawbacks when evaluating the potential clinical benefits of a medication. In the case of bempedoic acid, its advantage of not causing muscle-related symptoms is significant for patients' quality of life and treatment adherence. Further assessment of potential adverse effects is necessary to determine associated risks. What's more, a comprehensive consideration of the pros and cons, along with the evaluation of bempedoic acid's actual application in different patient populations, will help determine its value in clinical practice. As more information about the medication continues to emerge, healthcare professionals and patients can gain a better understanding of the potential of bempedoic acid, enabling more informed treatment decisions.

Our research possesses numerous strengths. Initially, the effectiveness of bempedoic acid was determined for enrolled patients, regardless of whether they had statin intolerance. In addition, we identified several noteworthy safety concerns, such as adverse events leading to discontinuation, gout, increased hepatic enzyme events, renal impairment events, and myocardial infarction.

A few limitations must be considered, however, with regard to this meta-analysis. To begin with, variations in patient baseline characteristics were observed among the studies that were included. Moreover, the duration of the present study was extremely brief, and long-term data on bempedoic acid requires improvement. An additional constraint of this meta-analysis pertains to publication bias in the included studies with respect to general efficacy results.

## Conclusion

Bempedoic acid is an efficacious and well-tolerated oral lipid-lowering agent that can be administered to patients who have or do not have statin intolerance. Further investigation is necessary to assess the clinical impact and safety for the long term.

## Supporting information

**S1 Checklist. PRISMA checklist.**
(DOCX)

**S1 Fig.** "trim and fill" technique of apolipoprotein B(A),non-high-density lipoprotein cholesterol(B) and total cholesterol(C).
(TIF)

**S2 Fig. Assessment of risk of bias in included studies.**
(TIF)

**S1 Table. Basic features of the included studies.**
(DOCX)

**S1 Data.**
(RAR)

## Acknowledgments

We would like to acknowledge the reviewers for their helpful comments on this paper.

## Author Contributions

**Conceptualization:** Yi Li.

**Data curation:** Jinghui Zhao.

**Formal analysis:** Yi Li.

**Funding acquisition:** Yi Li.

**Investigation:** Yi Li.

**Methodology:** Dan Hu.

**Project administration:** Yi Li.

**Resources:** Yi Li.

**Software:** Yi Li.

**Supervision:** Liqing Ma.

**Validation:** Yi Li.

**Visualization:** Yi Li.

**Writing – original draft:** Yi Li.

**Writing – review & editing:** Yi Li, Hongyu Gao.

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
