## [Decision Letter · Decision Letter 0]

13 Dec 2023

PONE-D-23-38386Safety and efficacy of bempedoic acid among patients with statin intolerance and those without: A meta-analysis and a systematic randomized controlled trial reviewPLOS ONE

Dear Dr. li,

Thank you for submitting your manuscript to PLOS ONE. After careful consideration, we feel that it has merit but does not fully meet PLOS ONE’s publication criteria as it currently stands. Therefore, we invite you to submit a revised version of the manuscript that addresses the points raised during the review process.

Please submit your revised manuscript by Jan 27 2024 11:59PM. If you will need more time than this to complete your revisions, please reply to this message or contact the journal office at plosone@plos.org. Please include the following items when submitting your revised manuscript:A rebuttal letter that responds to each point raised by the academic editor and reviewer(s). You should upload this letter as a separate file labeled 'Response to Reviewers'.A marked-up copy of your manuscript that highlights changes made to the original version. You should upload this as a separate file labeled 'Revised Manuscript with Track Changes'.An unmarked version of your revised paper without tracked changes. You should upload this as a separate file labeled 'Manuscript'.If applicable, we recommend that you deposit your laboratory protocols in protocols.io to enhance the reproducibility of your results. Protocols.io assigns your protocol its own identifier (DOI) so that it can be cited independently in the future. For instructions see: https://journals.plos.org/plosone/s/submission-guidelines#loc-laboratory-protocols. Additionally, PLOS ONE offers an option for publishing peer-reviewed Lab Protocol articles, which describe protocols hosted on protocols.io. Read more information on sharing protocols at https://plos.org/protocols?utm_medium=editorial-email&utm_source=authorletters&utm_campaign=protocols.

We look forward to receiving your revised manuscript.

Kind regards,

Hean Teik Ong

Academic Editor

PLOS ONE

Journal Requirements:

Additional Editor Comments:

Authors have produced a scientifically sound paper that can be accepted after minor revisions to address comments of the reviewers.

Reviewers' comments:

Reviewer's Responses to Questions

**Comments to the Author**

1. Is the manuscript technically sound, and do the data support the conclusions?

Reviewer #1: Yes

Reviewer #2: Yes

2. Has the statistical analysis been performed appropriately and rigorously? 

Reviewer #1: Yes

Reviewer #2: Yes

3. Have the authors made all data underlying the findings in their manuscript fully available?

Reviewer #1: Yes

Reviewer #2: Yes

4. Is the manuscript presented in an intelligible fashion and written in standard English?

Reviewer #1: Yes

Reviewer #2: Yes

5. Review Comments to the Author

Reviewer #1: The metanalysis although showed that Bempedoic acid is efficacious in reducing LDL cholesterol and no increase in the muscle associated occurrences there is drawback of drug discontinuation due to adverse effects, Authors should highlight the major adverse effects that lead to discontinuation of treatment. Authors should analyze the NET clinical benefit balancing the advantage of absence of muscle associated symptoms and downside of more adverse effects.

Reviewer #2: Good meta analysis on Bempedoic acid - a currently underutilized lipid lowering therapy.

Some minor observations are mainly on spelling and grammatical errors:

Line 129 should read "except" rather than "expect"

Line 236 should read "subgroup analyses report"

Line 243 should read "compared to" instead of "than"

Line 254 should read "requires"

6. PLOS authors have the option to publish the peer review history of their article (what does this mean?). If published, this will include your full peer review and any attached files.

Reviewer #1: No

Reviewer #2: No

---

## [Author Response · Author response to Decision Letter 0]

23 Dec 2023

Response to Reviewers

December 18, 2023 

Dear Editor, 

We would like to thank you and all reviewers for your insightful comments concerning the revisions of our manuscript entitled “Safety and efficacy of bempedoic acid among patients with statin intolerance and those without: A meta-analysis and a systematic randomized controlled trial review”. We have carefully reviewed the reviewers’ comments, and we have addressed all their questions/critiques/concerns point-by- point as below. All changes made to the text were tracked so that they may be easily identified. We believe that our manuscript has been significantly improved, and we hope it is now suitable for publication.

If you have any problems with our manuscript, please do not hesitate to contact us.

Journal Requirements:

Reply: Thank you, we have adjusted the format according to your suggestion

2.Did you know that depositing data in a repository is associated with up to a 25% citation advantage (https://doi.org/10.1371/journal.pone.0230416)? If you’ve not already done so, consider depositing your raw data in a repository to ensure your work is read, appreciated and cited by the largest possible audience. You’ll also earn an Accessible Data icon on your published paper if you deposit your data in any participating repository (https://plos.org/open-science/open-data/#accessible-data).

Reply: Thank you for your valuable advice, we will not consider this for the time being

Reply: Thank you, we have followed your suggestion. Thank you again

Reply: Thank you, we have followed your suggestion. Thank you again

Additional Editor Comments:

Authors have produced a scientifically sound paper that can be accepted after minor revisions to address comments of the reviewers.

Reply: Thank you for your valuable advice. We have carefully reviewed the reviewers’ comments, and we have addressed all their questions/critiques/concerns point-by- point as below. All changes made to the text were tracked so that they may be easily identified. We believe that our manuscript has been significantly improved, and we hope it is now suitable for publication.

Reviewers' comments:

Reviewer's Responses to Questions

Comments to the Author

1. Is the manuscript technically sound, and do the data support the conclusions?

Reviewer #1: Yes

Reviewer #2: Yes

Reply: Thank you for your affirmation, thank you

2. Has the statistical analysis been performed appropriately and rigorously? 

Reviewer #1: Yes

Reviewer #2: Yes

Reply: Thank you for your affirmation, thank you

3. Have the authors made all data underlying the findings in their manuscript fully available?

Reviewer #1: Yes

Reviewer #2: Yes

Reply: Thank you for your affirmation, thank you

4. Is the manuscript presented in an intelligible fashion and written in standard English?

Reviewer #1: Yes

Reviewer #2: Yes

Reply: Thank you for your affirmation, thank you

5. Review Comments to the Author

Reviewer #1: The metanalysis although showed that Bempedoic acid is efficacious in reducing LDL cholesterol and no increase in the muscle associated occurrences there is drawback of drug discontinuation due to adverse effects,

 Authors should highlight the major adverse effects that lead to discontinuation of treatment. 

Reply: Thank you for your valuable advice. We have highlighted the major adverse effects that lead to discontinuation of treatment in the discussion section.

Authors should analyze the NET clinical benefit balancing the advantage of absence of muscle associated symptoms and downside of more adverse effects.

Reply: We agree. When evaluating the clinical benefit of a medication like bempedoic acid, it is crucial for authors to analyze the net clinical benefit by weighing the advantage of the absence of muscle-associated symptoms against the potential downside of experiencing more adverse effects.By highlighting both the advantages and potential adverse effects, authors can provide a balanced assessment of the medication's clinical benefit.We have revised the discussion section as suggested.

Reviewer #2: Good meta analysis on Bempedoic acid - a currently underutilized lipid lowering therapy.

Reply: Thank you for your affirmation, thank you

Some minor observations are mainly on spelling and grammatical errors:

Line 129 should read "except" rather than "expect"

Line 236 should read "subgroup analyses report"

Line 243 should read "compared to" instead of "than"

Line 254 should read "requires"

Reply: Thanks for the reminder. We have revised the manuscript as suggested.

---

## [Decision Letter · Decision Letter 1]

15 Jan 2024

Safety and efficacy of bempedoic acid among patients with statin intolerance and those without: A meta-analysis and a systematic randomized controlled trial review

PONE-D-23-38386R1

Dear Dr. Yi Li,

We’re pleased to inform you that your manuscript has been judged scientifically suitable for publication and will be formally accepted for publication once it meets all outstanding technical requirements.

Kind regards,

Hean Teik Ong

Academic Editor

PLOS ONE

Additional Editor Comments (optional):

Reviewers' comments:

Reviewer's Responses to Questions

**Comments to the Author**

1. If the authors have adequately addressed your comments raised in a previous round of review and you feel that this manuscript is now acceptable for publication, you may indicate that here to bypass the “Comments to the Author” section, enter your conflict of interest statement in the “Confidential to Editor” section, and submit your "Accept" recommendation.

Reviewer #1: All comments have been addressed

Reviewer #2: All comments have been addressed

2. Is the manuscript technically sound, and do the data support the conclusions?

Reviewer #1: Yes

Reviewer #2: Yes

3. Has the statistical analysis been performed appropriately and rigorously? 

Reviewer #1: Yes

Reviewer #2: Yes

4. Have the authors made all data underlying the findings in their manuscript fully available?

Reviewer #1: Yes

Reviewer #2: Yes

5. Is the manuscript presented in an intelligible fashion and written in standard English?

Reviewer #1: Yes

Reviewer #2: Yes

6. Review Comments to the Author

Reviewer #1: My comments on the major side effects that lead to discontinuation of the medication and the issue of net clinical benefit of the drug have been addressed by the authors. No further comment.

Reviewer #2: All comments have been addressed, no further comments to add. A good review article on bempedoic acid.

7. PLOS authors have the option to publish the peer review history of their article (what does this mean?). If published, this will include your full peer review and any attached files.

Reviewer #1: No

Reviewer #2: No

---

## [Editor Report · Acceptance letter]

17 Jan 2024

PONE-D-23-38386R1 

PLOS ONE

Dear Dr. li, 

I'm pleased to inform you that your manuscript has been deemed suitable for publication in PLOS ONE. Congratulations! Your manuscript is now being handed over to our production team.

Kind regards, 

on behalf of

Dr. Hean Teik Ong 

Academic Editor

PLOS ONE